# LOCI-SEGMENTED:
# IMPROVING SCENE SEGMENTATION LEARNING

## ABSTRACT

Slot-oriented approaches for compositional scene segmentation from images and videos still depend on provided background information or slot assignments. We present Loci-Segmented (Loci-s) building on the slot-based location and identity tracking architecture Loci (Traub et al., ICLR 2023). Loci-s enables dynamic (i) background processing by means of a foreground identifying module and a background re-generator; (ii) top-down modified object-focused bottom-up processing; and (iii) depth estimate generation. We also improve automatic slot assignment via a slot-location-entity regularization mechanism and a prior segmentation network. The results reveal superior video decomposition performance in the MOVi datasets and in another established dataset collection targeting scene segmentation. Loci-s outperforms the state-of-the-art with respect to the intersection over union (IoU) score in the multi-object video dataset MOVi-E by a large margin and even without supervised slot assignments and without the provision of background information. We furthermore show that Loci-s generates well-interpretable latent representations. These representations may serve as a foundation-model-like interpretable basis for solving downstream tasks, such as grounding language, forming compositional rules, or solving one-shot reinforcement learning tasks.

## 1 INTRODUCTION

Visual scene understanding from images or videos presents unique challenges. Classical architectures such as CNNs (Liu et al., 2022) or ViTs (Vision Transformers) (Dosovitskiy et al., 2020) exacerbate existing limitations, being data-hungry, susceptible to adversarial attacks, and low on interpretability. To address these challenges, slot attention mechanisms have emerged as a promising avenue (Locatello et al., 2020). These architectures offer a way to bind features into 'slots' that dynamically represent distinct entities in a scene (Locatello et al., 2020), building upon prior work in attention mechanisms (Vaswani et al., 2017) and capsule networks (Sabour et al., 2017). Current state-of-the-art systems include the Slot Attention for Video model (SAVi++, Elsayed et al.), which however applies supervised slot assignments upon trial initialization, and the location and identity tracking slot-based recurrent architecture (Loci, Traub et al., 2023b), which assigns slots without supervision but relies on static backgrounds. Our work builds on Loci.

Loci has shown superior performance on the CATER challenge, in which objects (balls and cones) are transported hidden within other objects (cones). It is rather closely related to other slot-based object processing architectures including SAVi++ (Elsayed et al.; Locatello et al., 2020; Kipf et al., 2022; Wu et al., 2023), surveyed in (Yuan et al., 2023). It differs in (i) its slot-specific encoding approach that starts from pixels, (ii) its emergent disentanglement of objects from positions, and (iii) its event-oriented internal processing loop. Up to now, Loci relied on a static background-encoding module. As a result, it was not applicable to more complex dynamic backgrounds or moving cameras. Moreover, Loci was not able to profit from or predict depth information.

In this work, we enable Loci to deal with (i) dynamic complex backgrounds, (ii) videos where the camera is moving, and (iii) depth information. Additional incremental improvements enable the segmentation of scenes with more complex and diverse objects. Thereby, we enhance the state-of-the-art of scene segmentation algorithms in both the MOVi-* datasets (Greff et al., 2022) as well as in a scene segmentation dataset benchmark suite used in a recent review paper (Yuan et al., 2023). Our key contributions are:

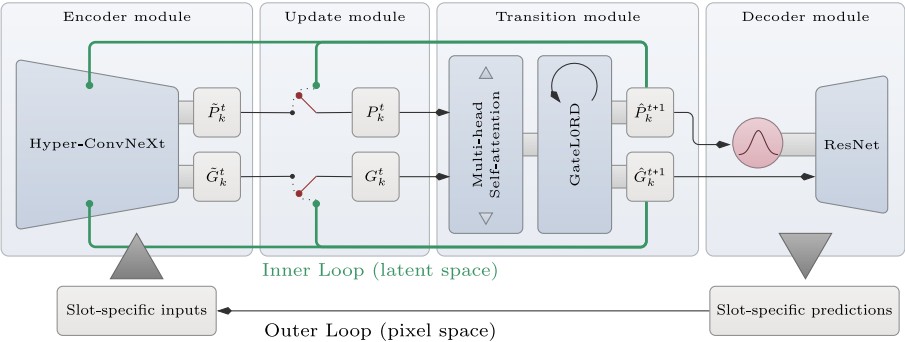

Figure 1: The primary Loci-s architecture features: a Hyper-ConvNeXt encoder, which generates Position and Gestalt codes slot-individually; an Update module, which adaptively fuses current encoder information with prior temporal predictions; a Transition module, which calculates object dynamics via a GateL0RD layer (i.e., a strongly gated RNN, cf. Gumbsch et al., 2021) and inter-slot interactions via self-attention; finally, a Decoder module, which computes sequential slot-wise predictions including depth estimates for the subsequent frame.

(i) A novel dynamic background processing module, which enables foreground-background separation including background depth and RGB reconstruction;

(ii) An enhanced encoder processing pipeline with top-down controlled second-order connections, facilitating adaptive object-focused encodings;

(iii) An enhanced decoder processing pipeline that introduces faster sequential mask-prioritized decoding and the generation of object depth estimates;

(iv) The optional inclusion of Scene-Relative-Depth as an input channel;

(v) The implementation of improved slot assignment techniques, including slot-location-entity regularization and the inclusion of a standard prior segmentation network;

(vi) Superior segmentation and video decomposition performance in numerous benchmarks.

## 2 LOCI ARCHITECTURE

Before detailing the novel extensions in Loci-s, we briefly introduce Loci (Traub et al., 2023b). Loci is a slot-based object-oriented processing architecture that consists of a slot-wise encoder, a transition, and a decoder module (cf., Figure 1).

**Encoder Module:** In contrast to other slot-based approaches (Yuan et al., 2023), Loci slots each start from the input image. At time point $t$, each slot $k$ receives as input the actual video frame $I^t$, the previous prediction error $E^t$, and a background mask $\hat{M}_{bg}^t$. Moreover, to focus each slot on its own object encoding, previous slot predictions are fed in as additional input, including its predicted position $\hat{Q}_k^t$ encoded as an isotropic Gaussian in pixel space, its visibility mask $\hat{M}_k^{t,v}$ and object mask $\hat{M}_k^{t,o}$ encoded as grayscale images, its RGB image $\hat{R}_k^t$, and the summed visibility masks of the remaining slots $\hat{M}_k^{t,s}$. As output, the encoder produces two key types of codes for each slot by means of a ResNet architecture: **Gestalt Codes** $\tilde{G}_k^t$: A 1D latent representations of an object's appearance, capturing shape, color, texture, and other visual attributes; **Position Codes** $\tilde{P}_k^t$: A disentangled spatial property code including the object's 2D location $(x_k, y_k)$, its size $(\sigma_k)$, and its distance in depth encoded as a priority code $(\rho_k)$.

**Transition Module:** The transition module contains a slot-wise recurrent module and a multi-head attention module. The recurrent module implements GateL0RD units, which encode LSTM-like recurrent cells with an even stronger shielding, to foster event-predictive encodings (Gumbsch et al., 2021). The multi-head attention module enables the object-interaction-oriented exchange of information between slots. As its result, the transition module outputs next object-respective Gestalt-Codes $\hat{G}_k^{t+1}$ and positions $\hat{P}_k^{t+1}$.

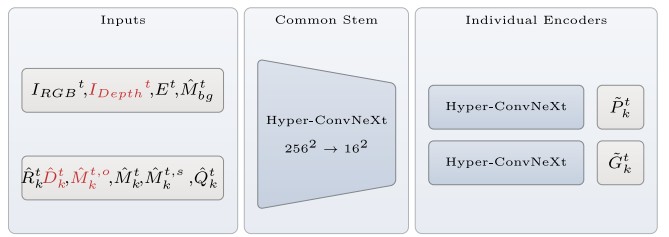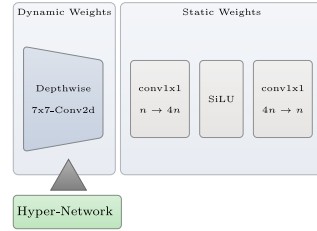

Figure 2: Left: Encoder visualization with the added depth information and slot-wise depth and object mask channels (in red). Right: A single Hyper-ConvNeXt block within the encoder where a top-down hyper-network translates $\hat{G}_k^t$ into spatial convolutional kernel weight residuals.

**Decoder Module:** The decoder module reconstructs the predicted scene starting from a 3D tensor that combines the Gestalt code vector as channels with the positional encoding $(x_k, y_k, \sigma_k)$. It then upscales this tensor to the full input resolution via a ResNet. The outputs are an RGB slot image $\hat{R}_k^{t+1}$, visibility mask $\hat{M}_k^{t+1,v}$, and position $\hat{Q}_k^{t+1}$. The scene is finally recomposed by combining the masked RGB outputs with respect to their respective priority codes $\hat{\rho}_k$ and the background mask.

## 3  METHODOLOGY

Loci-s builds on Loci but significantly enhances its abilities: We enable the processing and prediction of depth information; we design an even more dynamic encoder-decoder framework; and we introduce a dedicated background processing module. Detailed Loci-s network wiring and size information can be found in Appendix C.

### 3.1  DEPTH AS INPUT

We introduce a novel input channel to the Loci-s model, denoted as Scene-Relative Depth (see Appendix B for more details). Depth normalization is expected to significantly support object segmentation, as object edges will naturally be marked by spatial depth non-linearities. While this channel enhances performance further, our ablation studies below show that the Loci-s model yields superior performance in MOVi-E also without depth information as input.

### 3.2  ENCODER

The encoder and decoder subnetworks adopt a ConvNeXt-like architecture Liu et al. (2022), replacing the previously used ResNet architecture. Furthermore, we add an inner top-down processing loop to the architecture, which propagates predicted Gestalt code information $\hat{G}_k^t$ directly into the encoder. These codes are utilized within a hypernetwork to compute dynamic residuals for the depth-wise convolutional kernels present in the ConvNeXt blocks of the encoder (see Figure 2). This architectural modification enables the encoder to integrate top-down feedback into its computations, thereby improving the object-specific encoding pipeline.

### 3.3  DECODER

We introduce a cascaded decoder architecture, shown in Figure 3, and partition the Gestalt Code $G_k^t$ into three segments, each comprising 256 elements. These segments encode mask $Gm_k^t$, depth $Gd_k^t$, and RGB channels $Gr_k^t$. The Mask Decoder module uses element-wise multiplication between the Gestalt Code $\hat{Gm}_k^t$ and a two-dimensional isotropic Gaussian heatmap generated from Position Code $\hat{P}_k^t$. This modulated spatialized Gestalt Code is then subject to a compact convolutional neural network. The Depth Decoder module is implemented by a U-Net architecture. It decodes the depth information via the predicted Depth Gestalt Code $\hat{Gd}_k^t$, which is multiplied layer-wise with the computed object mask, enforcing masked outlines. The RGB Decoder module clones the Depth Decoder architecture but additionally receives the Depth Decoder's output as input. Finally, the RGB image of the encoded object is reconstructed via the RGB Gestalt code, which is layer-wise multiplied with the computed object mask and additionally informed by the generated depth estimations.

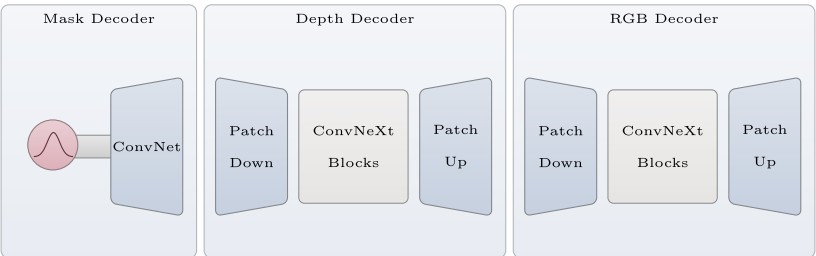

Figure 3: Decoder visualization illustrating the cascaded reconstruction strategy, first decoding the mask, then the depth, and finally the RGB image of a slot-encoded entity.

The cascading of the decoder architecture does not only encourage a disentangled encoding of an object's mask (i.e., its shape), its distance to the camera (i.e., its depth), and its appearance, but it also facilitates appearance reconstruction because the prediction of the mask is easier and then informs the depth and RGB-pattern reconstruction. Additionally, the cascaded decoder facilitates the reconstruction of the unoccluded raw mask $\hat{M}_k^{t,o}$ and the occlusion-aware mask $\hat{M}_k^t$, because only the Mask-decoder, but not the Depth and RGB decoders, needs to be re-run to generated the occlusion-aware mask.

## 3.4 BACKGROUND

Another pivotal enhancement in our work is the development of a Background Module, which is trained prior to the slotted architectural components. This module enables the application of Loci-s to environments with complex backgrounds, featuring both intricate backgrounds and moving cameras. As delineated in Figure 4, this module is bifurcated into two core elements: an Uncertainty Network and a Background Extraction Network.

The Uncertainty Network employs a U-Net architecture with ConvNeXt residual blocks. Skip connections between down-sampling and up-sampling layers avoid vanishing gradients. The network is trained in a supervised manner to compositionally segment the foreground in a scene, generating an uncertainty mask that predicts the provided foreground mask from either pure RGB or RGB+Depth depending on the used version (Loci-s or Loci-$s_d$). It thus learns to deem dynamic foreground objects 'uncertain', in contrast with the generally stable background elements in natural scenes.

The output from the Uncertainty Network serves as a masking function for the Background Extraction Network. This network implements a masked autoencoder using a Vision Transformer. This Vision Transformer is designed to predict both the RGB values and the depth map from either RGB alone or from both RGB and depth maps . By selectively masking-out foreground objects, we introduce a bias favoring the exclusive reconstruction of the background elements. To further accentuate this bias, we constrain the depth reconstruction module with a narrow bottleneck. The network is trained as a background autoencoder by masking the reconstruction loss with the inverse of the foreground mask.

## 3.5 PRETRAINING OBJECT ENCODINGS AND DECODINGS

The original Loci architecture was fully trained end-to-end without any information on objects whatsoever. While Loci-s could also be trained in this way, in order to speed-up learning and save computational resources, we implement a sequential pre-training strategy specifically tailored for Loci-s's encoder and decoder components. It is trained on single-object detection and reconstruction tasks. To initialize it, the encoder is furnished with an input frame with all slot-specific inputs nullified except for the slot-specific 2D Gaussian position, which is computed from the ground-truth target mask during this training stage. To avoid reliance on exact position encodings, the encoding is subjected to stochastic perturbations while ensuring its confinement within the mask's boundary. Any slot-specific inputs contingent with other slots are explicitly nullified. Note that during this pre-training stage we thus encourage slots to encode particular masks, but we do not initialize the slots explicitly to ground-truth bounding boxes. During all evaluations, we do not provide any supervised

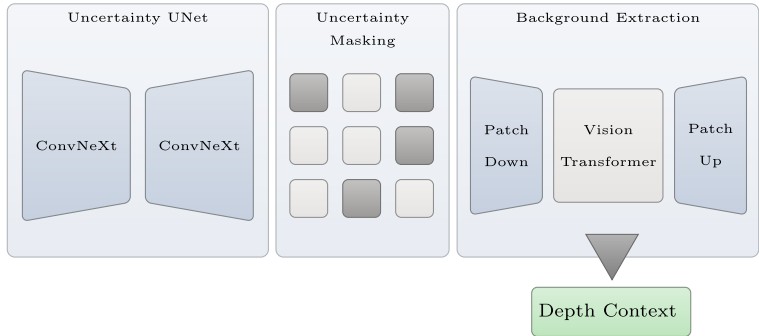

Figure 4: Background module: Input RGB or RGB+Depth is used to compute an Uncertainty or Foreground Mask via a U-Net. The mask is used in a Masked Autoencoder to reconstruct a background representation from the foreground masked input image (RGB or RGB+Depth).

slot information whatsoever. This is in stark contrast to SAVi++, where slots are initialized to the ground-truth bounding boxes in the first frame during training and evaluation (Elsayed et al.).

### 3.6 SEGMENTATION PREPROCESSING

In the process of pre-training the encoder-decoder architecture on discrete object instances, the Loci-s network acquires a foundational ability to en- and decode objects. Learning is furthermore supported by the pre-trained background module, which distinguishes foreground entities from the background context. The remaining challenge lies in the accurate identification and allocation of objects into distinct slots. SAVi++ provides ground-truth bounding box information to accomplish this step (Elsayed et al.). Our work explores three methodologies for the initial assignment of slots, without relying on ground-truth information.

First, we utilize a stochastic positioning strategy within the foreground mask that is generated by the Uncertainty Network. In particular, we sample a location uniformly randomly within the generated uncertainty map, which essentially predicts object masks. We initialize an empty slot with this location encoded as a 2D Gaussian position $Q_k^t$, similar to the pretraining of the object-specific encodings and decodings specified above. This serves as the most rudimentary technique.

This first approach, however, is susceptible to the erroneous partitioning of larger objects, because multiple random positions may be selected within the same object. To mitigate this, our second approach—termed "Regularized Initial Slots"—retains the random sampling paradigm during a warm-up phase. Following each network pass, though, we compute a similarity metric for each slot pair, based on both the Euclidean distance between their positional codes and the correlation of their Gestalt codes. Slots exhibiting a similarity below a predefined threshold are nullified in a stochastic manner.

The third approach employs a specialized segmentation network akin to YOLACT (Bolya et al., 2019), which was trained supervised using a Cross-Entropy loss comparing predicted instance masks with the best matching ground truth once. The best matching was computed using a linear sum assignment with IoU as the metric. For more details on the architecture or segmentation performance see appendix (cf. Table 7 and Table 3). Initial slot positions are calculated based on the instance masks outputted by this network. Additionally, the instance masks are used as a teacher forcing signal, instead of the actual slot masks, during a slot initialization phase that is applied on the initial frame.

## 4 EXPERIMENTS & RESULTS

In our experimentation pipeline, we initially pretrain our models on the Kubric MOVi-(a-f) dataset (Greff et al., 2022). Subsequently, we employ two distinct strategies: (1) full model training on MOVi-(a-e) for benchmarking against SAVi++ (Elsayed et al.), and (2) fine-tuning the encoder,

Table 1: Loci-s demonstrates largely superior performance in the MOVi Challenge, benchmarked against SAVi++ (Elsayed et al.). The results show that segmentation performance critically depends on the strategy for initial slot assignment. Note that SAVI++ provides ground-truth masks to each slot in the first frame. Loci-s, on the other hand, uses our novel segmentation preprocessing strategy. A segmentation network-informed slot assignment (seg) with depth information as input (Loci-$s_d$) yields the best score. Random slot assignments given the uncertainty map from the Uncertainty Module (rnd) clearly show the importance to start with good slot assignments, yielding performance worse than SAVi++ but still better than SAVi in MOVi-D and MOVi-E. Employing the regularized initial slot technique (reg) during the initialization phase yields intermediate outcomes. The smaller standard deviation ($\pm$) of our results also hints at a more reproducible training and evaluation of Loci-s than SAVI++ and especially SAVI.

| | mIoU↑ (%) | | | FG-ARI↑ (%) | | |
|---|---|---|---|---|---|---|
| Model | MOVi-C | MOVi-D | MOVi-E | MOVi-C | MOVi-D | MOVi-E |
| CRW | $27.8 \pm 0.2$ | $45.3 \pm 0.0$ | $47.5 \pm 0.1$ | * | * | * |
| SAVi | $43.1 \pm 0.7$ | $22.7 \pm 7.5$ | $30.7 \pm 4.9$ | $77.6 \pm 0.7$ | $59.6 \pm 6.7$ | $55.3 \pm 5.8$ |
| SAVi++ | $45.2 \pm 0.1$ | $48.3 \pm 0.5$ | $47.1 \pm 1.3$ | $\mathbf{81.9 \pm 0.2}$ | $\mathbf{86.0 \pm 0.3}$ | $84.1 \pm 0.9$ |
| Loci-$s_d$ (rnd) | $40.3 \pm 0.1$ | $40.9 \pm 0.3$ | $44.4 \pm 0.2$ | $58.5 \pm 0.6$ | $49.8 \pm 0.2$ | $60.8 \pm 0.2$ |
| Loci-$s_d$ (reg) | $45.7 \pm 0.2$ | $47.9 \pm 0.3$ | $49.2 \pm 0.3$ | $74.1 \pm 0.3$ | $74.0 \pm 0.9$ | $81.3 \pm 0.8$ |
| Loci-$s_d$ (seg) | $\mathbf{45.5 \pm 0.1}$ | $\mathbf{51.8 \pm 0.1}$ | $\mathbf{53.5 \pm 0.1}$ | $79.2 \pm 0.3$ | $81.1 \pm 0.2$ | $\mathbf{88.5 \pm 0.2}$ |
| Loci-s (rnd) | $33.4 \pm 0.3$ | $38.8 \pm 0.2$ | $41.2 \pm 0.2$ | $60.4 \pm 0.5$ | $54.3 \pm 0.3$ | $63.9 \pm 0.6$ |
| Loci-s (reg) | $35.7 \pm 0.2$ | $41.2 \pm 0.2$ | $42.4 \pm 0.1$ | $68.1 \pm 0.5$ | $71.4 \pm 0.7$ | $78.3 \pm 0.1$ |
| Loci-s (seg) | $36.2 \pm 0.1$ | $44.9 \pm 0.1$ | $47.0 \pm 0.1$ | $72.7 \pm 0.3$ | $79.5 \pm 0.2$ | $85.1 \pm 0.0$ |

decoder, and background modules on the datasets delineated in the Computational Scene Representation Review (Yuan et al., 2023) prior to Loci-s training on these datasets.

For video dataset training, we employ a warm-up phase comprising three iterations, during which only the encoder and decoder are updated, omitting the predictor. This warm-up occurs on the initial frame. We utilize truncated backpropagation through time (BPTT) with a sequence length of 2 for sequence-based learning. In contrast, for image datasets, we omit the warm-up phase and iteratively forward and backward propagate the same image for three cycles.

In the inference phase, we extend the warm-up iterations to 10 for both video and image data, which showed an empirical improvement in performance. Additionally, in image-centric tasks, we augment the number of full-architecture iterations to 10, totaling 20 processing steps: 10 for encoder/decoder-only warm-up and 10 for full architecture evaluation.

## 4.1 VIDEO EVALUATION

The design of Loci-s primarily centers around temporal predictions, hence a detailed comparative study is performed against SAVi++ on the MOVi-(c-e) dataset. To maintain evaluative consistency, we adhere to the same performance measures as outlined in the SAVi++ paper, namely the Per-Frame Intersection over Union (IoU) and the Per-Sequence Foreground Adjusted Rand Index (FG-ARI).

As illustrated in Table 1, Loci-s manifests a notable performance uplift, registering a 13.59% relative IoU improvement on the most demanding MOVi-E dataset, elevating the score from 47.1% (attained by SAVi++) to 53.5% (Loci-s with depth input and segmentation preprocessing). The FG-ARI metric, which encapsulates both spatial fidelity and the temporal consistency of the computed masks, presents a more nuanced landscape. While Loci-s achieves the best score in the MOVi-E dataset, SAVi++ exhibits superior performance in the MOVi-C and MOVi-D datasets. This performance discrepancy can be partially attributed to their architectural distinctions: SAVi++ incorporates an explicit history of previous frames to predict the current frame, while Loci-s fuses current observations to form the next frame prediction.

In light of these architectural design choices, Loci has demonstrated stable slot activations over extended temporal windows (Traub et al., 2023b). However, the masks decoded from the predicted Gestalt codes $M_k^{t+1}$ are less accurate than those derived directly from the encoder $\tilde{M}_k^t$. The encoder masks, on the other hand, suffer from low temporal consistency, since the information fusion of

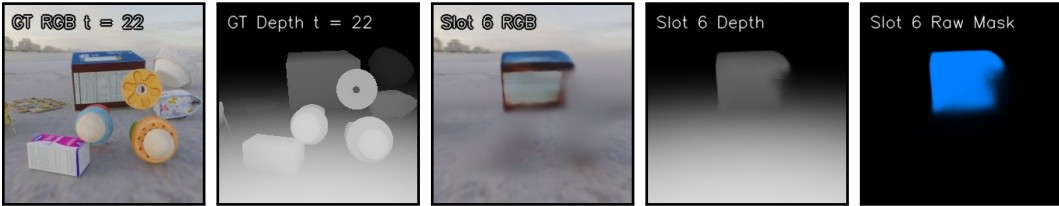

Figure 5: Example of an occluded object inference given ground truth RGB image and depth map.

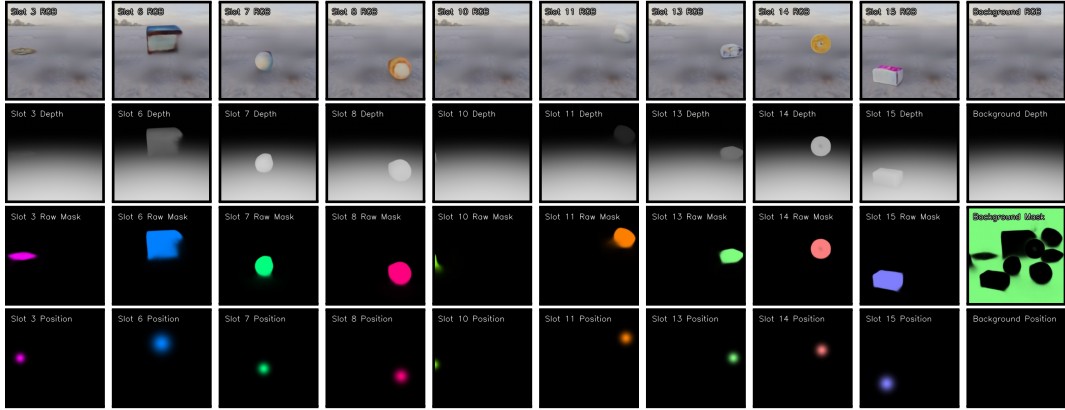

Figure 6: Interpretable slot-wise decomposition of the input from Figure 7 into RGB, depth, and mask reconstructions as well as position estimates (rows). Only occupied slots (columns) and the background module output (most right column) are shown.

$\tilde{G}_k^t$ with $G_k^{t-1}$ happens afterwards via the update gate and also inside the predictor itself (via the GateL0RD recurrences). At the moment the challenge remains to further improve the fusion of accurate mask reconstructions ($\tilde{G}_k^t$) with stable temporal predictions ($\tilde{G}_k^{t-1}$).

## 4.2 IMAGE EVALUATION

In a recent review paper about compositional scene understanding, Yuan et al. (2023) proposed a total of 6 datasets ranging in complexity from compositing MNIST to realistic texture simulations like MOVi-(c-e). These datasets are constructed in a way to perform two test: an in-distribution test with the same number of objects in a scene as seen during training (between 3 and 6); and another out-of-distribution test that probes generalization abilities with object numbers ranging from 7 to 10. In our experiments we used a pretraind (on MOVi-(a-f)) encoder-decoder network and fine-tuned it using all 6 datasets at once. We then further trained Loci-s on these combined 6 datasets and set the maximum number of slots to 6 during training. We then selected the model checkpoint form the epoch with the lowest validation error. For the generalization test we simply increase the maximum number of slots without further training to 10. We test the following metrics, which were reported by Yuan et al. (2023): Adjusted Mutual Information (AMI), Adjusted Rand Index (ARI), Intersection

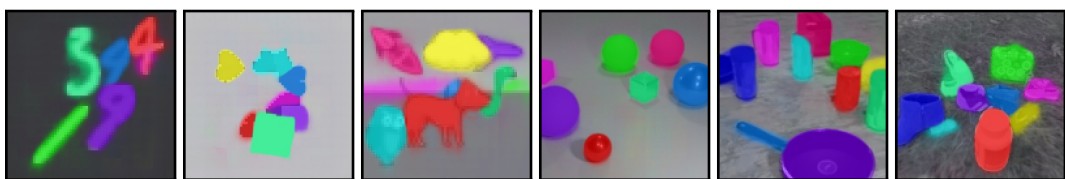

Figure 7: Loci-s segmentation example of the generalization datasets from the compositional scene understanding paper Yuan et al. (2023). The datasets are compositions of MNIST digits or dSprites, the Abstract Scene dataset, CLEVR, SHOP VRB, and a combination of GSO and HDRI-Haven.

Table 2: Loci-s largely outperforms all other reported approaches. Only in the object-specific adjusted mutual information (AMI) score and the object counting accuracy GMIOO is better.

| | AMI-A | ARI-A | AMI-O | ARI-O | IoU | F1 | OCA | OOA |
|---|---|---|---|---|---|---|---|---|
| Test 1 Validation (3-6 Objects) | | | | | | | | |
| AIR | 0.380 | 0.397 | 0.845 | 0.827 | N/A | N/A | 0.549 | 0.709 |
| N-EM | 0.208 | 0.233 | 0.341 | 0.282 | N/A | N/A | 0.013 | N/A |
| IODINE | 0.638 | 0.700 | 0.772 | 0.752 | N/A | N/A | 0.487 | N/A |
| GMIOO | 0.738 | 0.811 | **0.916** | 0.914 | 0.708 | 0.808 | **0.772** | **0.846** |
| MONet | 0.657 | 0.699 | 0.863 | 0.857 | N/A | N/A | 0.663 | 0.583 |
| GENESIS | 0.411 | 0.412 | 0.420 | 0.382 | 0.105 | 0.170 | 0.213 | 0.603 |
| SPACE | 0.640 | 0.678 | 0.817 | 0.765 | 0.630 | 0.739 | 0.436 | 0.666 |
| Slot Attention | 0.393 | 0.321 | 0.758 | 0.711 | N/A | N/A | 0.028 | N/A |
| EfficientMORL | 0.341 | 0.279 | 0.673 | 0.621 | N/A | N/A | 0.107 | N/A |
| GENESIS-V2 | 0.304 | 0.206 | 0.728 | 0.693 | N/A | N/A | 0.153 | 0.574 |
| Loci-s (rnd) | 0.835 | 0.918 | 0.735 | 0.891 | 0.742 | 0.822 | 0.421 | - |
| Loci-s (reg) | 0.838 | 0.921 | 0.730 | 0.898 | 0.731 | 0.810 | 0.443 | - |
| Loci-s (seg) | **0.844** | **0.922** | 0.748 | **0.920** | **0.781** | **0.860** | 0.505 | - |
| Test 2 Generalization (7-10 Objects) | | | | | | | | |
| AIR | 0.410 | 0.402 | 0.802 | 0.740 | N/A | N/A | 0.327 | 0.689 |
| N-EM | 0.256 | 0.268 | 0.354 | 0.261 | N/A | N/A | 0.017 | N/A |
| IODINE | 0.633 | 0.652 | 0.781 | 0.731 | N/A | N/A | 0.387 | N/A |
| GMIOO | 0.732 | 0.781 | **0.891** | 0.868 | 0.647 | 0.746 | **0.534** | **0.823** |
| MONet | 0.635 | 0.665 | 0.820 | 0.785 | N/A | N/A | 0.446 | 0.619 |
| GENESIS | 0.380 | 0.378 | 0.415 | 0.315 | 0.076 | 0.132 | 0.160 | 0.584 |
| SPACE | 0.628 | 0.639 | 0.802 | 0.717 | 0.543 | 0.654 | 0.265 | 0.650 |
| Slot Attention | 0.447 | 0.330 | 0.761 | 0.696 | N/A | N/A | 0.029 | N/A |
| EfficientMORL | 0.366 | 0.236 | 0.662 | 0.562 | N/A | N/A | 0.085 | N/A |
| GENESIS-V2 | 0.378 | 0.235 | 0.723 | 0.655 | N/A | N/A | 0.189 | 0.617 |
| Loci-s (rnd) | 0.820 | 0.866 | 0.766 | 0.875 | 0.667 | 0.755 | 0.228 | - |
| Loci-s (reg) | 0.828 | **0.888** | 0.768 | 0.865 | 0.637 | 0.724 | 0.244 | - |
| Loci-s (seg) | **0.832** | 0.877 | 0.783 | **0.905** | **0.706** | **0.792** | 0.315 | - |

over Union (IoU), F1 score and Object Counting Accuracy (OCA). As shown in Table 2 Loci-s shows superior performance in most metrics for both the in-distribution test and the generalization test. Note that we did not explicitly train or fine-tune Loci-s on each dataset individually, as don ine Yuan et al. (2023), but rather trained them on all datasets combined, scaling individual resolutions up to $256 \times 256$ where necessary. We expect even better performance with a dataset-specific fine-tuning of model and hyper parameters.

### 4.3 TOP DOWN FEEDBACK ABLATIONS

In Figure 8, we conduct a further ablation study to investigate the impact of top-down feedback in our architecture. We restrict our evaluation to pre-trained encoder-decoder networks, motivated by their substantially lower computational cost. Indeed, these networks are trainable using a single GTX 1080 GPU with a single slot for pre-training. Each experimental configuration is executed five times, utilizing a consistent set of five random seeds for reproducibility. Our ablation compares four scenarios evaluating the inner and outer top-down feedback loops: the gated inner loop flexibly controls the fusion of temporally predicted Position and Gestalt codes with the novel evidence from the new sensory input; the outer loop controls the hyper-network tuning the spatial convolutional encoder kernels top-down. We compare the performance of (i) the proposed architecture with both inner and outer feedback loops, (ii) a version with only outer feedback, (iii) a version with only inner feedback, and (iv) a baseline with no feedback mechanisms.

The results in Figure 8 demonstrate that the inclusion of top-down feedback is particularly advantageous for mask prediction tasks, resulting in a significant improvement in the Intersection-over-Union (IoU) metric. While the benefits for the Structural Similarity Index Measure (SSIM) in RGB

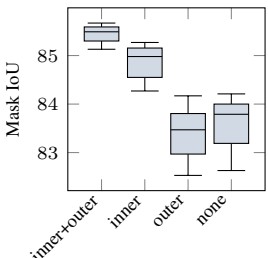 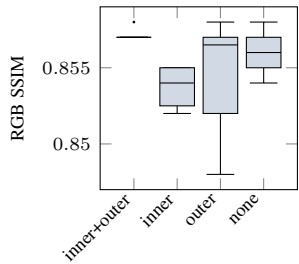 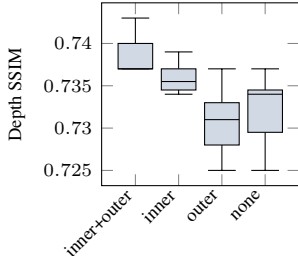

Figure 8: Ablating inner or outer feedback loop confirms their efficacy seeing improvements in per-object mask intersection-over-union (IoU) and depth structural similarity index measure (SSIM).

reconstruction are less consistent, the depth reconstruction task also profits from the presence of top-down feedback information.

## 5 CONCLUSION

The Loci-s model introduces several key innovations in the domain of scene understanding and object segmentation. A novel background reconstruction and foreground density estimation approach greatly facilitates object-oriented scene segmentations without relying on ground-truth slot initialization. Moreover, dynamic convolution kernels via a hyper-network-controlled top-down residual network facilitates object-specific visual encoding. Finally, the incorporation of depth information additionally facilitates segmentation performance event further. These advancements collectively contribute to a 13.59% relative improvement in IoU on the challenging MOVi-E dataset compared to state-of-the-art models like SAVi++. Even without depth information, Loci-s outperforms SAVI++ in the MOVi-E dataset, even though SAVi++ provides ground-trugh bounding boxes to initialize its slots in the first frame. Still, Loci-s falls short in some performance metrics, particularly in FG-ARI in the MOVi-C and MOVi-D datasets when compared to SAVi++. This suggests that while Loci-s excels in complex environments, it still struggles with fully accurate temporal object trackings. We suspect that this can be attributed to the fact that Loci-s fully compresses past video frame information in the internal recurrent state of its Transition Module. In contrast, SAVi++ maintains the full history applying attention-controlled fusion of all previous video frames. Our results demonstrate robustness in both in-distribution and out-of-distribution tests, highlighting the model's generalization abilities. However, it remains an open question whether this robustness extends to more varied or even more dynamic environments, and how it fares against models optimized for such scenarios. Furthermore, Loci-s shows great potential in terms of interpretability of deep learning systems, as shown in Figure 7 (further examples can be found in the appendix and supplementary video material).

Future work could integrate past frame information, as done in SAVi++, to enhance its temporal prediction abilities. Alternatively or additionally, the recurrent internal processing pipeline may be improved in future work. Taking inspiration from human cognition, from the binding problem, and from recent computational and conceptual insights into our modularized minds (Greff et al., 2020; Mattar & Lengyel, 2022; Heald et al., 2023; Butz et al., 2021; Schwöbel et al., 2021), the background processing module may yet be enhanced to a universal background extraction module relative to which foreground objects may be extracted. Furthermore, optimally distributing cognitive processing resources onto currently task-relevant objects and interactions remains as an important challenge. We believe that segmentation-oriented algorithms, such as Loci-s, constitute one crucial foundation-model-like module that offers itself to be effectively combined with (i) reinforcement learning, planning, and reasoning approaches and (ii) language processing modules in future work.

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

## A  APPENDIX

### A.1  CLOSING THE INNER LOOP

We enhance Loci's object tracking abilities similar to Traub et al. (2023a), which draws inspiration from Kalman filtering (Kalman, 1960). Originally, Loci predicts the next object states via a pixel space-routed outer loop (see Figure 1; outer loop). We draw inspiration from work in model-based reinforcement learning, which has recently advocated latent world model predictions (Hafner et al., 2019; 2020; Ha & Schmidhuber, 2018; Schrittwieser et al., 2020). These allow the imagination of future scene dynamics via an inner loop, without explicit pixel-based generations. Similarly, we apply an inner processing loop in Loci-s. (see Figure 1; inner loop).

In accordance with Kalman filtering, Loci-s is enabled to linearly interpolate between the current sensor information and its predictions. Formally, the current object states $S_k^t = (G_k^t, P_k^t)$ become a linear blending of the observed object states $\tilde{G}_k^t, \tilde{P}_k^t$ and the predicted object states $\hat{G}_k^t, \hat{P}_k^t$:

$$G_k^t = \alpha_k^{t,G} \tilde{G}_k^t + (1 - \alpha_k^{t,G})\hat{G}_k^t \tag{1}$$

$$P_k^t = \alpha_k^{t,P} \tilde{P}_k^t + (1 - \alpha_k^{t,P})\hat{P}_k^t \tag{2}$$

The weighting $\alpha$ is specific for each Gestalt and position code in each slot $k$. Importantly, Loci-s learns to regulate this percept gate on its own in a fully self-supervised manner. It learns an update function $g_\theta$, which takes as input the observed state $\tilde{S}_k^t$, the predicted state $\hat{S}_k^t$, and the last positional encoding $P_k^{t-1}$:

$$(z_k^{t,G}, z_k^{t,P}) = g_\theta(\tilde{S}_k^t, \hat{S}_k^t, P_k^{t-1}) + \varepsilon \qquad \text{with} \quad \varepsilon \sim \mathcal{N}(0, \Sigma), \tag{3}$$

Table 3: Performance of our segmentation prepossessing network. While achieving adequate performance on evaluation datasets, the preprocessing network clearly fails on the generalization dataset.

| mIoU↑ (%) | | | | |
|---|---|---|---|---|
| MOVi-C | MOVi-D | MOVi-E | Review datasets Test 1 | Review datasets Test 2 |
| $88.39 \pm 0.03$ | $82.48 \pm 0.05$ | $80.89 \pm 0.07$ | $90.98 \pm 0.12$ | $67.31 \pm 0.05$ |

We model $g_\theta$ with a feed-forward network. To be able to fully rely on its own predictions, Loci-s needs to be able to fully close the gate by setting $\alpha$ exactly to zero. We therefore use a rectified hyperbolic tangent to compute $\alpha$:

$$(\alpha_k^{t,G}, \alpha_k^{t,P}) = \max(0, \tanh((z_k^{t,G}, z_k^{t,P}))). \tag{4}$$

An $L_0$ loss on gate openings encourages the reliance on internal beliefs rather than external updates.

## B  DEPTH INPUT NORMALIZATION

We log-normalized the Scene-Relative Depth, according to Equation 5:

$$d = \frac{1}{1 + \exp\left(\frac{\hat{d}-\mu}{\sigma}\right)}, \tag{5}$$

where $\hat{d}$ represents the natural logarithm of the raw depth values. Parameters $\mu$ and $\sigma$ denote the mean and standard deviation of the log-transformed depth, respectively.

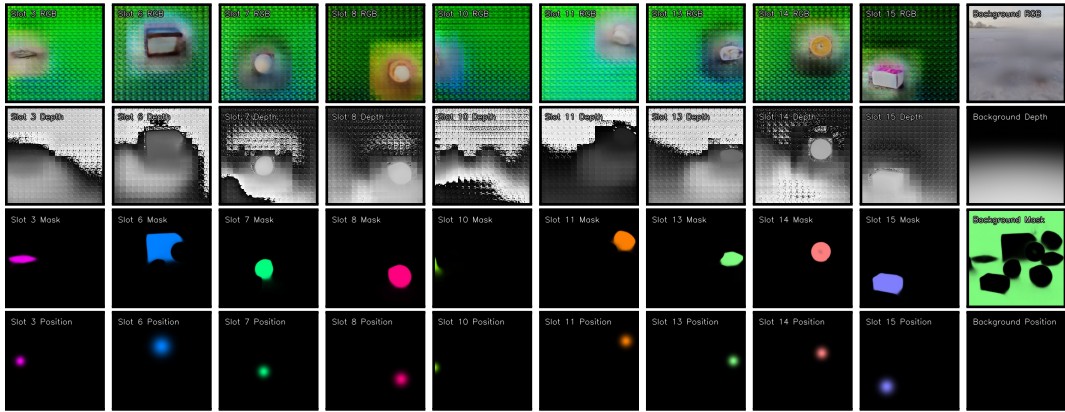

Figure 9: The slotwise decomposition of the input from Figure 7 into unmasked rgb and depth reconstructions per slot and the background rgb and depth reconstruction. For simplicity we only show occupied slots.

## C  DETAILED LOCI-S SIZE AND WIRING INFORMATION

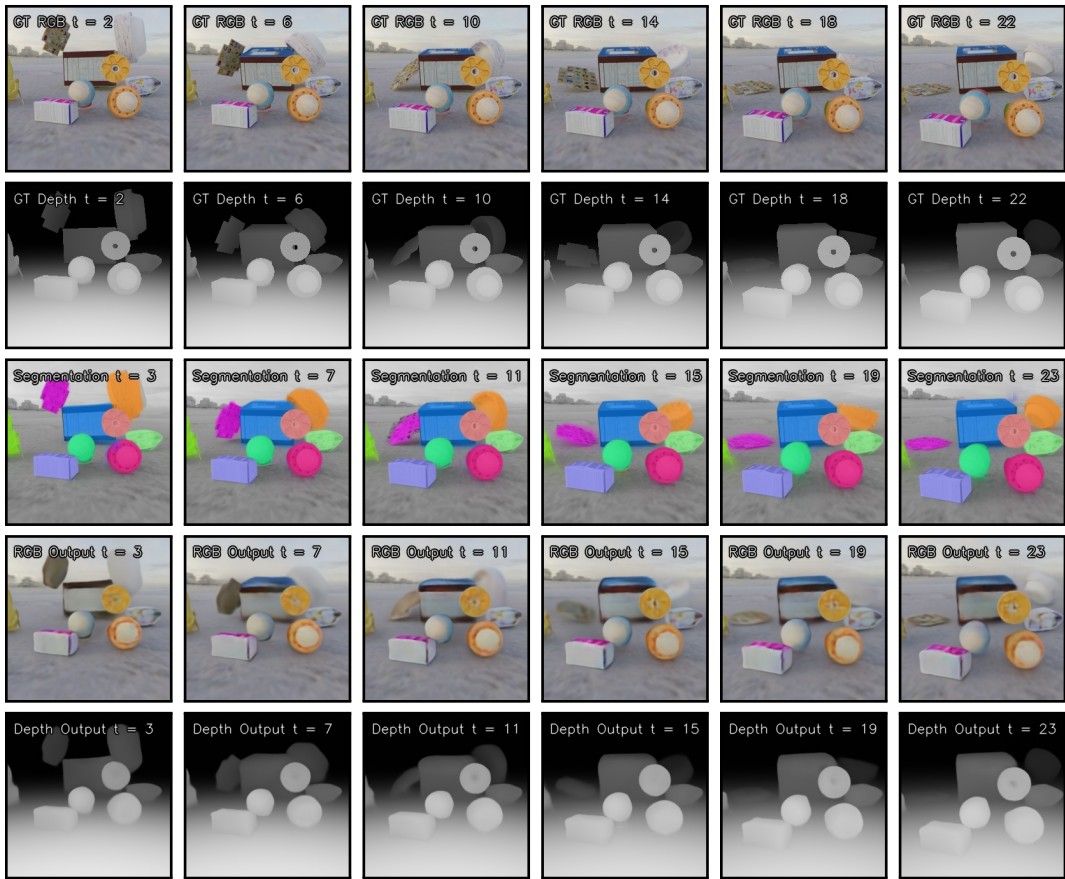

Figure 10: Qualitative Analysis of Results on the MOVi-E Dataset: The top two rows show the input frames while the third row shows the target frame superimposed with the slot masks, and the last two rows show the next frame predictions.

Table 4: Encoder Architecture, for more information see the source file in *nn/hyper_encoder.py*

| Component | Layer | Configuration |
|---|---|---|
| Encoder Base | ResidualPatchEmbedding | Conv2D(16, 32, 4, 4) + AvgPool + Channel Copy |
| | HyperConvNext | $32 \rightarrow 32$ |
| | ResidualPatchEmbedding | Conv2D(32, 64, 2, 2) + AvgPool + Channel Copy |
| | HyperConvNext | $64 \rightarrow 64$ |
| | ResidualPatchEmbedding | Conv2D(64, 128, 2, 2) + AvgPool + Channel Copy |
| | HyperConvNext | $128 \rightarrow 128$ |
| | HyperConvNext | $128 \rightarrow 128$ |
| | HyperConvNext | $128 \rightarrow 128$ |
| Position Encoder | HyperConvNext | $128 \rightarrow 128$ |
| | HyperConvNext | $128 \rightarrow 128$ |
| | HyperConvNext | $128 \rightarrow 4$ |
| | FeaturesMapToPosition | |
| Gestalt Base Encoder | HyperConvNext | $128 \rightarrow 128$ |
| | HyperConvNext | $128 \rightarrow 128$ |
| Mask Gestalt Encoder | ResidualPatchEmbedding | Conv2D(128, 256, 2, 2) + AvgPool + Channel Copy |
| | HyperConvNext | $256 \rightarrow 256$ |
| | HyperConvNext | $256 \rightarrow 256$ |
| | HyperConvNext | $256 \rightarrow 256$ |
| | HyperConvNext | $256 \rightarrow 256$ |
| | PositionPooling | |
| Depth Gestalt Encoder | ResidualPatchEmbedding | Conv2D(128, 256, 2, 2) + AvgPool + Channel Copy |
| | HyperConvNext | $256 \rightarrow 256$ |
| | HyperConvNext | $256 \rightarrow 256$ |
| | HyperConvNext | $256 \rightarrow 256$ |
| | HyperConvNext | $256 \rightarrow 256$ |
| | PositionPooling | |
| RGB Gestalt Encoder | ResidualPatchEmbedding | Conv2D(128, 256, 2, 2) + AvgPool + Channel Copy |
| | HyperConvNext | $256 \rightarrow 256$ |
| | HyperConvNext | $256 \rightarrow 256$ |
| | HyperConvNext | $256 \rightarrow 256$ |
| | HyperConvNext | $256 \rightarrow 256$ |
| | PositionPooling | |

Table 5: Predictor Architecture, for more information see the source file in *nn/predictor.py*

| Component | Layer | Configuration |
|---|---|---|
| UpdateController | Linear | $1550 \rightarrow 256$ |
| | SiLU | |
| | Linear | $256 \rightarrow 256$ |
| | SiLU | |
| | Linear | $256 \rightarrow 2$ |
| Predictor | InputEmbedding | |
| |   Linear | $774 \rightarrow 1024$ |
| |   SiLU | |
| |   Linear | $1024 \rightarrow 1024$ |
| | GateL0rd | $1024 \rightarrow 1024$ |
| | MultiheadSelfAttention | $1024 \rightarrow 1204$ |
| | GateL0rd | $1024 \rightarrow 1024$ |
| | MultiheadSelfAttention | $1024 \rightarrow 1204$ |
| | GateL0rd | $1024 \rightarrow 1024$ |
| | MultiheadSelfAttention | $1024 \rightarrow 1204$ |
| | GateL0rd | $1024 \rightarrow 1024$ |
| | MultiheadSelfAttention | $1024 \rightarrow 1204$ |
| | GateL0rd | $1024 \rightarrow 1024$ |
| | OutputEmbedding | |
| |   Linear | $1024 \rightarrow 1024$ |
| |   SiLU | |
| |   Linear | $1024 \rightarrow 774$ |

Table 6: Decoder Architecture, for more information see the source file in *nn/decoder.py*

| Component | Layer | Configuration |
|---|---|---|
| MaskDecoder | GestaltPositionFussion | |
| | Conv2d | $256 \to 128$, kernel=3, pad=1 |
| | SiLU | |
| | Conv2d | $128 \to 64$, kernel=3, pad=1 |
| | SiLU | |
| | Conv2d | $64 \to 32$, kernel=3, pad=1 |
| | SiLU | |
| | Conv2d | $32 \to 128$, kernel=1 |
| | TransposedConv2d | $128 \to 1$, kernel=16, stride=16 |
| DepthDecoder | MaskEncoder | |
| | Conv2d | $1 \to 128$, kernel=16, stride=16 |
| | SiLU | |
| | Conv2d | $128 \to 32$, kernel=1 |
| | GestaltMaskFussion | Gestalt * MaxPool(mask, kernel=16) |
| | Concat | ModulatedGestalt, EncodedMask, PositionalEmbedding |
| | Conv2d | $304 \to 64$, kernel=1 |
| | ConvNeXt | $64 \to 64$ |
| | ConvNeXt | $64 \to 64$ |
| | ConvNeXt | $64 \to 64$ |
| | Conv2d | $64 \to 256$, kernel=1 |
| | SiLU | |
| | TransposedConv2d | $256 \to 1$, kernel=16, stride=16 |
| RGBDecoder | MaskEncoder | |
| | Conv2d | $1 \to 128$, kernel=16, stride=16 |
| | SiLU | |
| | Conv2d | $128 \to 32$, kernel=1 |
| | DepthEncoder | |
| | Conv2d | $1 \to 256$, kernel=16, stride=16 |
| | SiLU | |
| | Conv2d | $256 \to 64$, kernel=1 |
| | GestaltMaskFussion | Gestalt * MaxPool(mask, 16) |
| | Concat | ModulatedGestalt, EncodedMask, EncodedDepth, PositionalEmbedding |
| | Conv2d | $368 \to 128$, kernel=1 |
| | ConvNeXt | $128 \to 128$ |
| | ConvNeXt | $128 \to 128$ |
| | ConvNeXt | $128 \to 128$ |
| | ConvNeXt | $128 \to 128$ |
| | ConvNeXt | $128 \to 128$ |
| | Conv2d | $128 \to 512$, kernel=1 |
| | SiLU | |
| | TransposedConv2d | $512 \to 3$, kernel=16, stride=16 |

Table 7: Segmentation preprocessing, see *nn/proposal_v2.py* for more details

| Component | Layer | Configuration |
|---|---|---|
| | Cat | Depth + 2D Grid(-1,1) |
| | ResidualPatchEmbedding | Conv2D(3, 64, 4, 4) + AvgPool + Channel Copy |
| | ConvNeXt | 64 → 64 |
| | ResidualPatchEmbedding | Conv2D(64, 128, 2, 2) + AvgPool + Channel Copy |
| | ConvNeXt | 128 → 128 |
| | ConvNeXt | 128 → 128 |
| | ResidualPatchEmbedding | Conv2D(128, 256, 2, 2) + AvgPool + Channel Copy |
| | ConvNeXt | 256 → 256 |
| | ConvNeXt | 256 → 256 |
| | ConvNeXt | 256 → 256 |
| | ResidualPatchEmbedding | Conv2D(256, 512, 2, 2) + AvgPool + Channel Copy |
| | ConvNeXt | 256 → 256 |
| | HyperNetwork | |
| | GlobalAvgPool | |
| | Linear | 512 → 512 |
| | SiLU | |
| | Linear | 512 → 512 |
| | SiLU | |
| | Linear | 512 → 512 |
| | ConvNeXt | 512 → 512 |
| SegmentationUNet | Conv2d | 512 → 2048, kernel=1 |
| | SiLU | |
| | Conv2d | 2048 → 256, kernel=2, stride=2 |
| | ConcatFeatures | |
| | Conv2d | 512 → 256, kernel=1 |
| | ConvNeXt | 256 → 256 |
| | Conv2d | 256 → 1024, kernel=1 |
| | SiLU | |
| | Conv2d | 1024 → 128, kernel=2, stride=2 |
| | ConcatFeatures | |
| | Conv2d | 256 → 128, kernel=1 |
| | ConvNeXt | 128 → 128 |
| | Conv2d | 128 → 512, kernel=1 |
| | SiLU | |
| | Conv2d | 512 → 64, 2, stride=2 |
| | ConcatFeatures | |
| | Conv2d | 128 → 64, kernel=1 |
| | ConvNeXt | 64 → 64 |
| | Conv2d | 64 → 512, kernel=1 |
| | SiLU | |
| | Conv2d | 512 → 32, kernel=4, stride=4 |
| | ApplyHyperWeights | Features @ weights, 32, 16 |

Table 8: Background Architecture, for more information see the source file in *nn/decoder.py*

| Component | Layer | Configuration |
|---|---|---|
| Uncertainty UNet | MaskEncoder | |
| | ResidualPatchEmbedding | Conv2D(4, 16, 4, 4) + AvgPool + Channel Copy |
| | ConvNeXt | $16 \rightarrow 16$ |
| | ResidualPatchEmbedding | Conv2D(16, 32, 2, 2) + AvgPool + Channel Copy |
| | ConvNeXt | $32 \rightarrow 32$ |
| | ResidualPatchEmbedding | Conv2D(32, 64, 2, 2) + AvgPool + Channel Copy |
| | ConvNeXt | $64 \rightarrow 64$ |
| | ResidualPatchEmbedding | Conv2D(64, 128, 2, 2) + AvgPool + Channel Copy |
| | ConvNeXt | $128 \rightarrow 128$ |
| | ConvNeXt | $128 \rightarrow 128$ |
| | ResidualPatchUpscaling | Conv2D(128, 64, 2, 2) + Upscale + Channel Avg |
| | ConvNeXt | $64 \rightarrow 64$ |
| | ResidualPatchUpscaling | Conv2D(64, 32, 2, 2) + Upscale + Channel Avg |
| | ConvNeXt | $32 \rightarrow 32$ |
| | ResidualPatchUpscaling | Conv2D(32, 16, 2, 2) + Upscale + Channel Avg |
| | ConvNeXt | $16 \rightarrow 16$ |
| | ResidualPatchUpscaling | Conv2D(16, 1, 4, 4) + Upscale + Channel Avg |
| Background Extractor Base-Encoder | PatchEmbedding | |
| |   Conv2d | $4 \rightarrow 256$, kernel=16, stride=16 |
| |   SiLU | |
| |   Conv2d | $256 \rightarrow 64$, kernel=1 |
| | MHA-Layer | $64 \rightarrow 64$ |
| | MHA-Layer | $64 \rightarrow 64$ |
| Background Extractor RGB-Encoder | MHA-Layer | $64 \rightarrow 64$ |
| Background Extractor Depth-Encoder | MHA-Layer | $64 \rightarrow 64$ |
| | Bottleneck | token avg + cross attention to single token |
| | Sigmoid | |
| | Binarize | $x \leftarrow x + x(1 - x)\mathcal{N}(0, 1)$ |
| Background Extractor Depth-Decoder | ConvNeXt | $64 \rightarrow 64$ |
| | ConvNeXt | $64 \rightarrow 64$ |
| | PatchUpscaling | |
| |   Conv2d | $64 \rightarrow 256$, kernel=1 |
| |   SiLU | |
| |   TransposedConv2d | $256 \rightarrow 1$, kernel=16, stride=16 |
| Background Extractor RGB-Decoder | Depth-Encoder | |
| |   PatchEmbedding | $1 \rightarrow 64$ |
| |   ConvNeXt | $64 \rightarrow 64$ |
| | Cross-Attention-Layer | $64 \rightarrow 64$ |
| | ConvNeXt | $64 \rightarrow 64$ |
| | ConvNeXt | $64 \rightarrow 64$ |
| | PatchUpscaling | |
| |   Conv2d | $64 \rightarrow 256$, kernel=1 |
| |   SiLU | |
| |   TransposedConv2d | $256 \rightarrow 1$, kernel=16, stride=16 |

