# OpenReview forum: "Loci-Segmented: Improving Scene Segmentation Learning"
_ICLR.cc/2024/Conference — Submitted to ICLR 2024_

### Official Review · Reviewer_wWut · 2023-10-31

**Soundness:** 3 good
**Presentation:** 3 good
**Contribution:** 3 good
**Rating:** 5
**Confidence:** 3

**Summary:**

This paper proposes Loci-s (Loci-Segmented) to tackle the problem of slot-oriented scene representation. Loci-s is an extension of the Loci architecture with structure change, additional inputs and etc. The proposed methods shows state-of-the-art performance on the challenging MOVi-E dataset, demonstrating its ability to deal with complex environments.

**Strengths:**

1)This paper extends the Loci model to Loci-s with several innovations.
2)The advancements in Loci-s collectively contribute to a 32.06% relative improvement in IoU on the challenging MOVi-E dataset compared to state-of-the-art models like SAVi++.

**Weaknesses:**

1)Since the proposed method is built upon Loci, I think there should be more comparisons between Loci and Loci-s in the experimental section.

2)The authors mentioned that instead of the residual structure used in Loci, the encoder and decoder subnetworks in Loci-s have been revamped to adopt a ConvNeXt-like architecture. I wonder how much performance improvement is brought by this structure change.

3)“The third methodology involves the deployment of a specialized segmentation network akin to YOLACT (Bolya et al., 2019). ” How is this segmentation network trained, and what is its performance?

4)The proposed method incorporates some additional input information for performance boost, e.g. the segmentation input (seg), depth map (sd). How much is the time cost?

5)(NOT IMPORTANT) Seems there is a missing reference (shown as ?) in sentence “Loci is rather closely related to other slot-based object processing architectures (Elsayedet al.; ?; Kipf et al., 2022; Wu et al., 2023)...”

**Questions:**

Same as weakness.

---

> ### Author Response · Authors · 2023-11-21
> **Answer to Reviewer wWut**
>
> Please also refer to our reply to all reviews for a clarification of the key merits of our work.
>
> Thank you for your insightful comments. Here are our replies:
>
> 1) The original Loci cannot be trained with dynamic backgrounds (it uses a Gaussian-Mixture-Model to extract one single background for training- and test set) and therefore a direct comparison is not applicable.
>
> 2) ConvNeXt has proven to be the superior residual CNN by the original paper (A ConvNet for the 2020s) we therefore adapted this superior design since it also nicely fits with our Hyper-Convolution approach. Although it would certainly be an interesting ablation we kindly ask the reviewer to consider that it might be out of scope (and energy wasteful) for us to re-ablate already established designs.
>
> 3) The segmentation network was trained supervised using a Cross-Entropy loss with an optimal instance mask assignment (computed with the Hungarian algorithm on the IoU). We now also detail this in the methods and report the segmentation accuracy in the appendix.
>
> 4) There is no real additional cost besides training the additional segmentation network. All methods (rnd), (reg) and (seg) are used in a warm-up phase where the first frame is shown during 10 cycles that involve only the encoder and decoder. The computational cost of the segmentation network is comparable to one additional slot in Loci-s.

---

### Official Review · Reviewer_xtqd · 2023-10-31

**Soundness:** 3 good
**Presentation:** 3 good
**Contribution:** 3 good
**Rating:** 6
**Confidence:** 1

**Summary:**

This work focuses on the compositional scene representation and proposes a scene segmentation neural network based on the previous model named Loci. To build their model, they extend Loci with three modifications, including a pre-trained dynamic background
module, a hyper-convolution encoder module, and a cascaded decoder module. Extensive experiments conducted on the MOVi dataset show the effectiveness of the proposed method. Besides, the proposed method can generate well-interpretable latent representations and may serve as a foundation-model-like interpretable basis for solving downstream tasks.

**Strengths:**

1. Good performance. The proposed method achieves good performance on the MOVi dataset.
2. The proposed can generate well-interpretable latent representations, which is helpful in building interpretable foundation models.

**Weaknesses:**

To ACs and authors: I am not an expert in this field and cannot find any strong reasons to reject this work. Please refer to other reviewers' comments for rebuttal and decision.

**Questions:**

None.

---

> ### Author Response · Authors · 2023-11-21
> **Answer to Reviewer xtqd**
>
> Thank you for your encouraging comments. Please refer to our replies to all reviewers for a detailed summary of further improvements and the key highlights of our work for your convenience.

---

### Official Review · Reviewer_czo6 · 2023-11-01

**Soundness:** 2 fair
**Presentation:** 1 poor
**Contribution:** 2 fair
**Rating:** 3
**Confidence:** 3

**Summary:**

This paper proposes an architecture for unsupervised scene segmentation given RGB or RGBD video input. The method builds on the "Loci" paper, but re-designs many components and adds in a pre-trained foreground/background segmentation model. The main result is that this combination of changes greatly improves results, both qualitatively and quantitatively.

**Strengths:**

Quantitatively the method here clearly outperforms prior work (on the mIOU metric) in the the MOVI-* datasets.

**Weaknesses:**

Overall this paper is very difficult to follow. The "Loci" method, on which this is based, is never quite made clear on its own, and then every subsequent section makes big changes to the architecture without much motivation, and without a connecting story or high-level idea.

The section on the "Background Module" never mentions this, but the abstract and the section on "Segmentation Preprocessing" describe the background module as "pre-trained", apparently for a segmentation task that "distinguishes foreground entities from the background context". My guess is that much of the performance gain is coming from this.

I have a variety of smaller questions which the authors may like to answer, but overall it seems to me that this paper needs a very heavy rewrite.

**Questions:**

What are Gestalt codes?

What are the two predictions about object positions? The paper says "we introduce a dedicated background processing module that generates both predictions about object positions as pixel densities".

The paper mentions using something called "GateL0RD units" but these are never really described.

The paper says that the "Gestalt codes are binarized to create an information bottleneck that fosters the development of factorized compositional entity encodings." I am unclear on why binarization will make the representation compositional.

Section 2.1 focuses on improving Loci's "object tracking abilities", but the earlier section (describing Loci) never mentioned any object tracking happening, and tracking is never mentioned again. What is the idea here?

The paper mentions that the decoder "reconstructs the predicted scene via slot-wise density maps as object masks." What are slot-wise density maps?

The paper briefly mentions an "L0 loss on gate openings" but it is not clear what ground truth is used for this loss. Is it maybe just a regularization term, penalizing the L0 norm?

Section 2.2 introduces a depth input and an equation to normalize it, but it is not clear where this fits with the inner loop described in the previous section.

For Table 2 it would be great to clarify what dataset these experiments happen in, and what the metrics are.

---

> ### Author Response · Authors · 2023-11-21
> **Answer to Reviewer czo6**
>
> Please also refer to our reply to all reviews for a clarification of the key merits of our work.
>
> Thank you for your insightful comments. Here are our replies:
>
>  - We polished the overall presentation (see answer to all reviewers) and apologize for the mediocre accessibility of some of the novel components in the previous version of the paper.
>
>  - The Background module indeed is trained supervised using  ground truth foreground masks. We now clarify this in the methods. Clearly, slot attention methods like Loci greatly benefit from an accurate background module.
>
>  - Gestalt-Codes are a compressed latent vector that encodes position invariant object properties needed to reconstruct the object at any location determined by the Position-Code. We also make this more clear now in our section on the (base) Loci Architecture.
>
>  - We clarified the sentence about "pixel densities". It should now be much clearer that the Uncertainty Network as part of the Background Module computes a foreground mask, which is learned supervised.
>
>  - GateL0RD units are LSTM-like recurrent cells with an even stronger shielding. We now further clarify this in the paper.
>
>  - The binarization of Gestalt codes was introduced in the original Loci paper as a strong bottleneck that outperformed a variational one, but since it is not really relevant for the understanding of Loci-s we removed this sentence.
>
>  - The main objective of the original Loci paper was object tracking trough occlusions, but since Loci-s focuses on object segmentation we removed the statements regarding object tracking.
>
>  - Slot-wise density maps are the instance segmentation masks computed by each slot. We clarify this now.
>
>  - The L0 loss on gate openings is a regularization loss that punishes any gate values except zero with the same gradient. But since this inner loop is not relevant for the segmentation task we moved it to the appendix. This also puts the depth input into a more consistent story line.
>
>  - We added descriptions for the datasets and metrics from the compositional scene understanding paper.

---

### Official Review · Reviewer_vymZ · 2023-11-05

**Soundness:** 2 fair
**Presentation:** 2 fair
**Contribution:** 2 fair
**Rating:** 5
**Confidence:** 3

**Summary:**

The paper extends the location and identity tracking architecture Loci to scene segmentation by adding a pre-trained dynamic background
module, a hyper-convolution encoder module, and a cascaded decoder module. The proposed method and each components are validated to be effectiveness by extensive experiments.

**Strengths:**

The experiments are extensive.

**Weaknesses:**

1. The work is a little incremental, compared to Loci, so that its novelty is slim.
2. The principle and motivation of the proposed modules are not clearly explained.

**Questions:**

see weakness

---

> ### Author Response · Authors · 2023-11-21
> **Answer to Reviewer vymZ**
>
> Please refer to our objection with respect to the incremental nature of our work in the general reply.
>
> We agree that we did not motivate the novel models sufficiently well and have done so now (see paper blue passages). To summarize:
> - Background Module: Necessary to effectively distinguish complex dynamic backgrounds from foreground objects, addressing a key limitation in the original Loci system, which only worked if the background was static for the whole training and test set.
> - Hyper-Convolution Encoder: To integrate top-down feedback, enhancing object-specific encoding and improving bottom-up processing. We have an ablation in the appendix that shows a significant improvement in IoU and depth reconstruction for Hyper-Convolutions.
> - Cascaded Decoder: To sequentially generate object masks, depth maps, and RGB reconstructions, leading to more accurate and detailed segmentations.
>
> These modules are integral to the enhanced performance of Loci-s in complex scene segmentation, marking a critical step beyond the capabilities of the original Loci architecture.

---

### Author Response · Authors · 2023-11-21
**Answer to all Reviewers**

Thank you for your consideration and time.
Here we provide answers to your common questions and address other concerns in individual answers.
We apologize for having failed to accessibly highlight the functionality and novelty of our model and hope that the revision succeeds in doing so.

In brief:
 - All our changes are marked in blue in the revision
 - We adjusted the introduction to highlight our main novel contributions and made them more clear
  - We extended and improved the introduction of the base Loci architecture and its components.
  - We also improved the writing of the methods section.
  - We motivate the individual novel modules and components in more detail.
  - Our work it not incremental, as we
    - introduce a completely novel background and foreground segregation technique,
    - a novel hyper-network encoder pipeline enabling slots to process and focus on more complex and diverse objects,
    - a novel cascaded decoder pipeline, enabling faster mask generations, depth predictions and more complex RGB object image generation;
    - processing depth as input (optionally) and predicting a depth map of the background and the individual objects.
  - As our main results, we beat the state-of-the-art in most evaluation metrics in recent image and video segmentation benchmarks.

Please note that we unfortunately had a mismatch in the way SAVI++ computed the IoU metric. Therefore, we had to rerun all our MOVi evaluations yielding slightly worse results, which are nonetheless still largely outperforming SAVi++.
In this respect, we also noted that SAVi++ actually provides ground-truth slot information upon slot initialization (first frame) - besides providing the full history of frames.
This gives SAVi++ indeed yet another huge advantage, which we were not aware of before.
As a result, the fact that we mostly beat SAVi++ is even more significant.
We now emphasize this very important point throughout the paper.

---

### Meta-Review · Area_Chair_QEnE · 2023-12-12

**Metareview:**

This paper extends the slot-based location and identity tracking architecture Loci from a single static background to dynamic background.  The proposed model consists of the addition of a pre-trained dynamic background module, a hyper-convolution encoder module, and a cascaded decoder module.

The strengths of the paper are extensive experimentation and sizable performance gains.  The weaknesses of the paper are incremental novelty, confusing writing, and insufficient experimental validation.

The paper has received 4 reviews, with ratings 6/5/5/3.  Authors have provided detailed rebuttals and updated the submission significantly.  The only positive review (6) only has confidence level 1, with the reviewer explicitly asking to consult other reviews.

Through the clarification of rebuttals, it becomes clear that the proposed model is supervised with ground truth foreground-background masks that the baselines do not have (in order to learn object segmentation without supervision), which most likely is the source of large performance gains.  Reviewers find such experimental settings not comparable (or unfair to baselines), and experimental validation insufficient.  It would be helpful to tease out the contribution from the extra foreground-background supervision and the additional modules designed over Loci.

The AC agrees with the consensus of negative reviews and recommends rejection.

**Justification For Why Not Higher Score:**

Incremental novelty, confusing writing, and insufficient experimental validation.
Negative review consensus.

**Justification For Why Not Lower Score:**

N/A

---

### Decision · Program_Chairs · 2024-01-16

Reject